# Immunotherapy in Bladder Cancer: Current Methods and Future Perspectives

**DOI:** 10.3390/cancers12051181

**Published:** 2020-05-07

**Authors:** Mikołaj Wołącewicz, Rafał Hrynkiewicz, Ewelina Grywalska, Tomasz Suchojad, Tomasz Leksowski, Jacek Roliński, Paulina Niedźwiedzka-Rystwej

**Affiliations:** 1Institute of Biology, University of Szczecin, 71-412 Szczecin, Poland; mikolaj.wolacewicz@gmail.com (M.W.); rafal.hrynkiewicz@gmail.com (R.H.); 2Department of Clinical Immunology and Immunotherapy, Medical University of Lublin, 20-093 Lublin, Poland; jacek_rolinski@gmail.com; 3St. John’s Cancer Centre, 20-090 Lublin, Poland; 4Department of Urology, Regional Specialist Hospital, 26-060 Czerwona Góra, Poland; tritomek@gmail.com (T.S.); t.lex@onet.eu (T.L.)

**Keywords:** bladder cancer, immunotherapy, checkpoint inhibitor

## Abstract

Bladder cancer is one of the most significant genitourinary cancer, causing high morbidity and mortality in a great number of patients. Over the years, various treatment methods for this type of cancer have been developed. The most common is the highly efficient method using Bacillus Calmette-Guerin, giving a successful effect in a high percentage of patients. However, due to the genetic instability of bladder cancer, together with individual needs of patients, the search for different therapy methods is ongoing. Immune checkpoints are cell surface molecules influencing the immune response and decreasing the strength of the immune response. Among those checkpoints, the PD-1 (programmed cell death protein-1)/PD-L1 (programmed cell death protein ligand 1) inhibitors aim at blocking those molecules, which results in T cell activation, and in bladder cancer the use of Atezolizumab, Avelumab, Durvalumab, Nivolumab, and Pembrolizumab has been described. The inhibition of another pivotal immune checkpoint, CTLA-4 (cytotoxic T cell antigen), may result in the mobilization of the immune system against bladder cancer and, among anti-CTLA-4 antibodies, the use of Ipilimumab and Tremelimumab has been discussed. Moreover, several different approaches to successful bladder cancer treatment exists, such as the use of ganciclovir and mTOR (mammalian target of rapamycin) kinase inhibitors, IL-12 (interleukin-12) and COX-2 (cyclooxygenase-2). The use of gene therapies and the disruption of different signaling pathways are currently being investigated. Research suggests that the combination of several methods increases treatment efficiency and the positive outcome in individual.

## 1. Introduction

Bladder cancer (BC) is the sixth most commonly diagnosed cancer in men worldwide and the 10th when considering men and women together [1]. The worldwide age-standardized incidence rate (per 100,000 person/years) is 9.6 for men and 2.4 for women [1]. In Europe, the overall age-standardized incidence rate is 20.2 for men and 4.3 for women. Greece has the highest age-standardized incidence rate of all European countries (40.4 in men and 4.5 in women) and Austria has the lowest (9.9 in men and 3.0 in women) [1]. Approximately 550,000 new cases of BC were diagnosed worldwide in 2018, with 200,000 deaths [1]. The incidence rate of BC has increased in many European countries, although mortality rates have declined in more developed regions. With population aging and growth, the absolute incidence of BC might continue to increase in European nations [2].

Smoking is the most significant risk factor for BC. It is linked to 50–65% of cases in men and 20–30% of cases in women. The incidence of BC is directly related to the duration of smoking and the number of cigarettes smoked per day [2,3]. Occupational factors are considered the second most important risk factor for BC [3]. Workers exposed to aromatic amines, polycyclic aromatic hydrocarbons, tobacco and tobacco smoke, combustion products, and heavy metals are at an increased risk [4].

Urothelial carcinoma originating in the bladder is the most common histological type of cancer. It is diagnosed in a nonmuscle invasive stage in more than 70% of cases, requiring only minimally invasive, local treatment. Unfortunately, the disease has a high rate of recurrence and treatment may have to be administered more than once. In contrast, muscle-invasive and metastatic stages of the disease need multimodal treatment strategies, including surgical treatment and chemotherapy in neoadjuvant, adjuvant, or palliative settings [5].

Cancer treatment methods that modify the immune status have a prominent place in oncology in recent years. Immunotherapy is usually used to complement conventional cancer treatments such as surgery, chemotherapy, and radiotherapy. For some cancers, immunotherapies are used as first-line treatment [6]. Immunotherapy in cancer treatment is a method that involves the patient’s immune system to modify or increase defense mechanisms against a developing cancer [6,7]. The first clinical application of immunotherapy was recorded in the 1890s, when William Coley first used a bacterial preparation called Coley toxin. The effect of clinical trials was small. The toxin provided an early demonstration of the potential to produce an antitumor response by using the patient’s immune system [6]. It was not until the mid-20th century that immunotherapies gained importance as part of standard cancer treatment, although they showed significant toxicity. Therapies were associated with the beginning of cell therapy with the development of bone marrow transplantation by Fritz Bach et al. in the 1960s and the production, testing, and approval in clinical trials of a high dose of IL-2 (interleukin 2) for metastatic renal and melanoma in the 1990s [8,9]. Currently, several types of immunotherapy are used to treat cancer, including immune checkpoint inhibitors, T-cell transfer therapy, monoclonal antibodies, treatment vaccines, and immune system modulators [7].

Research for best-tailored treatment for BC is ongoing, and immunotherapy seems to be the most promising prospect.

## 2. Bacillus Calmette–Guerin (BCG)

Bacillus Calmette–Guerin (BCG) is a weakened strain of *Mycobacterium bovis*. However, according to the European Association of Urology, there are currently 10 strains used for BCG therapy, but none of them has shown superiority over the others [10].

BCG use as cancer treatment was investigated in an animal model in 1974 [11], and in 1976 the first report on the successful use of BCG in BC was published [12]. In 1980 Lamm et al. reported that the use of BCG therapy following transurethral resection of bladder tumors (TURBT) reduces the chance of relapse compared to patients receiving only TURBT [13,14].

Currently, intravesical therapy with BCG is standard practice in the treatment of nonmuscle invasive BC (NMIBC), including in situ cancer, high-grade papillary tumors, and invasive plaque-proprious tumors [15]. Noninvasive tumors account for 70–80% of BC cases [16,17]. The standard treatment for this type of cancer is TURBT, followed by intravesical treatment with BCG or chemotherapy, as described by Lamm et al. years ago [14]. Whether BCG or chemotherapy is used depends on the progression and recurrence of the disease [18,19,20].

Although the BCG vaccine has been used to treat BC for decades, its mechanism of action is not yet fully understood [15,21]. BC cells themselves may play a role involving the attachment and internalization of BCG, the presentation of BCG and cancer antigens to cells of the immune system, and the mass release of cytokines and chemokines that occurs during BCG therapy [22,23]. What is certain is that BCG causes a strong innate immune response that leads to long-term adaptive immunity [21,24]. BCG therapy elicits an inflammatory reaction involving different immune cell subsets that kill cancer cells by direct cytotoxicity or by the secretion of toxic compounds, like the tumor necrosis factor-inducing ligand. The immune cell subsets that may be involved include CD4+ and CD8+ lymphocytes, NK (natural killer) cells, granulocytes, macrophages, and dendritic cells. Some cancer cells are also killed directly by the BCG [15].

T lymphocytes are present in the inflammatory infiltrate in the bladder of BCG-treated patients [25], and there is evidence that natural killer cells are cytotoxic against BCG-infected BC cells [26]. Granulocytes are also present in the inflammatory infiltrate in the bladder. In the mouse model, they were necessary for a proper immune response [27,28], along with CD4+ and CD8+ T cells [29]. Furthermore, BCG-exposed dendritic cells may stimulate the cytotoxicity of T lymphocytes against BCG-infected cancer cells [30]. Finally, macrophages are another component of the inflammatory infiltrate in the bladder of BCG-treated patients, and they are cytotoxic against cancer cells when stimulated by BCG [31,32].

BCG immunotherapy gives a high percentage of positive response, which is 55–65% for high-risk papillary tumors and 70–75% for carcinoma in situ (CIS) [33,34,35,36]. Unfortunately, as many as 25–45% of patients will not benefit from BCG therapy. In addition, about 40% of patients have relapse despite initial successes with BCG [21]. Despite the constant development of medicine and technology, the percentage of patients in whom BCG therapy does not have a positive effect remains similar to that reported in the early 1990s (30–35% of patients remain resistant to this method of treatment) [31,32]. Currently, patients can be divided into three groups: BCG resistant, BCG relapse, or BCG intolerance [33,37]. The differences between these groups remain extremely important because they can provide information on the response of individual patients to BCG therapy. Many studies are underway, including those showing promising early results in selected patients who do not respond to BCG but long-term results are still distant [33].

For this reason, understanding the immunobiology of BCG-induced tumor immunity is necessary to tailor BCG treatment to specific patients and to improve efficacy as well as to reduce intolerance to this therapy.

While some researchers are still trying to refine the BCG treatment method, some teams have focused on other, equally promising, immunotherapies against bladder cancer.

## 3. Checkpoints’ Inhibitors Pathway

Responsiveness to checkpoint inhibitors (mainly PD-1/PD-L1 (programmed cell death protein-1/programmed cell death protein ligand 1) and CTLA4 (cytotoxic T cell antigen)) is the key to successful cancer therapy, but still not every patient achieves clinical benefit [38]. Immune checkpoint efficacy is affected by various factors, among which tumor genomics, host germline genetics, PD-L1 levels, and gut microbiome may be enumerated [38]. Generally, in tumors, mutated or incorrectly expressed proteins are processed via the immunoproteasome into peptides that are usually loaded onto MHC (major histocompatibility complex) class I molecules, which further not always are able to elicit CD8^+^ T cell response [38]. This may lead to generating MHC-presented immunogenic neoepitopes [38]. It was shown, that once the intratumor heterogeneity rises, neoantigen-expressing clones became more homogenous with the differential expression of PD-L1 [39].

Also, mutations on several signaling pathways may influence the effectiveness of immune checkpoints inhibitors [39]. This was confirmed in bladder cancer for Janus kinase (JAK) signaling pathway, where negative regulator of JAK-SOCS3 (suppressor of cytokine signaling 3) was reduced, with simultaneous upregulation of miR-221 (micro RNA), leading to enhanced cell apoptosis, and attenuated cell proliferation [40]. Such dysregulation of JAK (Janus activated kinase)-STAT (signal transducer and activator of transcription) signaling pathway was also confirmed in patients with bladder cancer with high STAT3 expression [41].

Influence of the BC on cell cycle was also noticed, while some proteins like DEPDC1 (DEP domain containing 1) and MPHOSPH1 (M-phase phosphoprotein 1) are usually overexpressed in bladder cancer cells [42].

Overall, bladder cancer is a genetically heterogenous disease, with a high rate of somatic mutations, including genes involved in cell-cycle regulation, chromatin regulation, and signaling pathways [43]. In bladder cancer, according to The Cancer Genome Atlas (TCGA) Research Network, mutations of genes not significantly mutating in any other type of cancer were noticed [43]. Among most frequent mutations, one can enumerate aneuploidy, chromosomal instability, and fractional allelic losses [44]. Thus, those differences in the molecular features of BC, together with personal characteristic of patients, may seriously influence the efficacy of the use of the immune checkpoint inhibitors.

### 3.1. PD-1/PD-L1

Programmed cell death protein 1 (PD-1) and its ligands, programmed death ligand 1 (PD-L1) and 2 (PD-L2) [11], are part of an immune system checkpoint, which negatively regulates the immune system to weaken its response to antigens. PD-1 is expressed on the surface of activated T and B lymphocytes and macrophages (Figure 1) and PD-L1 is highly expressed by antigen-presenting cells [45]. PD-1 binding to PD-L1 blocks the activation of T cells, thereby reducing the production of IL-2 (interleukin-2) and interferon gamma [45]. This promotes self-tolerance by preventing the immune system from indiscriminately attacking the cells of the body, but it can also stop the immune system from attacking cancer cells that express PD-L1. PD-1/PD-L1 inhibitors are antibodies that block either of these two molecules, cancelling the checkpoint activity and thus resulting in T cell activation [46]. They were first introduced as second-line therapy in BC treatment, but are slowly establishing themselves as first-line therapy [47]. There are currently three PD-L1 inhibitors and two PD-1 inhibitors approved by the FDA (Food and Drug Administration) for BC treatment (Table 1).

#### 3.1.1. Atezolizumab

Atezolizumab, a humanized monoclonal antibody of isotype IgG1, [11,54,55] was the first PD-1/PD-L1 checkpoint inhibitor approved by the FDA and accepted by the European Association of Urology (EUA) [10] as second-line therapy for patients with advanced BC [56]. It is recommended for patients with advanced or metastatic disease, in whom treatment with platinum derivatives has been ineffective and patients need to be PD-1 positive [10,57].

#### 3.1.2. Avelumab

Avelumab is also an IgG1 antibody directed against PD-L1. It was approved only by the FDA in 2017 for urothelial carcinoma [11,58].

A phase Ib trial was carried out with patients with advanced or metastatic urothelial cancer, following platinum-based chemotherapy [59]. An objective response rate of 18.2% was demonstrated, including five complete and three partial responses. The median survival was 13.7 months, and the annual survival rate was 54.3%. Side effects such as fatigue, weakness, nausea, and infusion-related reactions were observed [59].

The results of a phase II trial showed that avelumab induced a persistent antitumor response in patients with advanced urothelial cancer whose tumors progressed during or after platinum-based chemotherapy [60].

#### 3.1.3. Durvalumab

Durvalumab is a monoclonal IgG1k antibody approved only by the FDA in 2017 for the treatment of BC [61]. Studies in phases I and II patients have confirmed the effectiveness of durvalumab: 20.4% of patients showed an immune response to the drug, and 4.9% showed a complete response [62]. Lately, research was conducted on the use of durvalumab in patients with BC and it was concluded that the activity of this particular drug is high in PD-L1 positive and negative patients, but higher responses were noted in patients with high PD-L1 expression [63]. An interesting study was held to find out whether the use of circulating tumor DNA (ctDNA) may influence the effect of durvalumab [64] and the results show that changes in somatic mutations in ctDNA are correlated with this anti-PD-L1 form of therapy and may be a good predictor of a successful immunotherapy.

#### 3.1.4. Nivolumab

Nivolumab is a human monoclonal antibody of type IgG4, approved by the FDA for advanced BC in 2017 [65] and by the EUA on the basis of a single-arm phase trial, including 270 platinum pretreated patients, receiving objective response rate nearly 20% [10].

A phase 1/2 study was conducted in patients with advanced or metastatic BC. A median overall survival of 9.7 months was recorded, with an annual survival of 46%. There was no significant difference among patients with PD-L1 overexpression. In addition, 59% of patients reported adverse drug reactions and 10% experienced serious adverse events [66].

A phase II clinical trial involving patients receiving platinum-based chemotherapy showed a two-month progression-free period and a median survival of 8.74 months. However, a difference in drug effects was observed in patients with PD-L1 overexpression compared to patients in the low-expression subgroup. Furthermore, 64.6% of patients reported adverse events, of which 17.8% were serious (grade 3 or 4) [51].

#### 3.1.5. Pembrolizumab

Pembrolizumab is a humanized IgG4/kappa monoclonal antibody used in the treatment of various types of cancer and approved by the FDA and EAU for BC, based on the ongoing phase III trials [10,11,67]. According to the European Medicines Agency (EMA), this drug may be used as a first-line therapy in urothelial cancer [68].

All the documented studies available to date in patients with advanced BC indicate the positive effects of pembrolizumab [69,70]. Bellmunt et al. compared the effects of pembrolizumab, classic chemotherapy, docetaxel, paclitaxel, and vinflunine [69]. The use of pembrolizumab resulted in a longer mean survival of approximately three months compared to paclitaxel, docetaxel, and vinflunine. A better safety profile compared with the other drugs was also noted. A relationship was observed between the efficacy of the drug and smoking, which may suggest a high mutational load in smokers [71]. Additionally, ongoing studies on patients with advanced urothelial BC are investigating a combined treatment involving pembrolizumab and colony-stimulating factor 1 receptor-a cell-surface molecule expressed on monocytes/macrophages [72]. Macrophages exist in a continuum of polarization states ranging from the inflammatory M1 phenotype to the pro-tumorigenic M2 phenotype. The presence of M2 macrophages in urothelial BC stroma often means BCG immunotherapy failure [72]. The newest report shows [73] that even though this drug may induce immune-related adverse effects, such as myocarditis and myasthenia gravis, this treatment may be safely and successfully used. Moreover, pembrolizumab was confirmed to be a rational therapy for patients with defects in DNA repairing [74].

Pembrolizumab is the first immune-checkpoint inhibitor that showed significantly better overall survival than chemotherapy [11].

### 3.2. Anti-CTLA-4 Antibodies

Cytotoxic T lymphocyte-associated protein 4 (CTLA-4) is a surface molecule expressed by activated T cells. CTLA-4 binds B7.1 and B7.2 ligands, which are expressed on B lymphocytes, dendritic cells, and macrophages [11]. CTLA-4 is a co-stimulant necessary for the activation of T lymphocytes [11,75,76]. It negatively regulates the immune response, but its mechanism of action is not yet fully understood. However, because it is structurally related to CD28, one suggestion is that it competes with CD28 in terms of ligand binding. Another suggestion involves the direct inhibition of CTLA-4 cytoplasmic tail signaling [77,78,79].

The inhibition of CTLA-4 may increase the regulation of the immune response to BC. This is the underlying hypothesis behind ongoing research into anti-CTLA-4 antibodies that are to be used as single agents in BC treatment (Figure 1).

Vaccine S-288310 may be used as an alternative. It works by activating cytotoxic T cells. Clinical studies show that this method is highly effective, and the vaccine was well tolerated by patients. The criterion for the use of vaccine S-288310 was increased expression of the HLA-A 24:02 (human leukocyte antigen) gene in patients [54,80].

#### 3.2.1. Ipilimumab

Ipilimumab is a monoclonal anti-CTLA-4 antibody developed by Bristol–Meyers Squibb and used for the treatment of melanoma. It is also used in combination with nivolumab for the treatment of advanced renal cell carcinoma and of different types of metastatic colorectal cancer [81].

Carthon et al. investigated the safety of the antibody in patients with BC and found that ipilimumab had a tolerable safety profile. Most of the adverse events were grade 1 or 2, and two patients had their surgery postponed because of immune-related adverse events [82].

Studies on combining ipilimumab with PD-1/PD-L1 blockers are in progress [83], but further research is needed on the effect of this drug in patients with BC.

#### 3.2.2. Tremelimumab

The use of tremelimumab, a humanized monoclonal anti-CTLA-4 antibody, is being investigated in combination with durvalumab [83]. However, Tremelimumab has not yet been approved by the FDA for cancer treatment. In a study conducted on 18 patients with advanced BC, most of the side effects were mild, but one patient died of intestinal perforation [84]. More research is needed on the way this drug works and its effects on patients. No data exist on this particular drug according to EAU guidelines [10].

### 3.3. Combined Therapies

In the treatment of many types of cancer, not only in bladder cancer, so-called combined therapies are used, i.e., a combination of two or more methods of treatment [10,85]. At present, the use of combined therapies is an important element and remains under the strict interest of scientists. Some combinations of therapies enable better effects not only in understanding the patient’s response to the drug but also in reducing the recurrence of cancer.

The main method in the treatment of CIS tumors is the use of TURB, also in combination with intravesical injections. The combination of Mitomycin C (MMC) and BCG is also widely used in the treatment of NMIBC (non-muscle-invasive bladder cancer) [86]. Researchers confirmed that the combination of these two methods had a positive effect on reducing relapses, but was more toxic compared to BCG monotherapy [86]. Thus, it was found that the combination of BCG and MMC alone is not better than BCG therapy alone [87]. A study conducted with a combination of BCG and IFN (interferon)-2α in patients at high risk of relapse and progression did not show a clear difference in relapse and progression over BCG-only therapy. In contrast, the use of MMC weekly and then monthly BCG for a change from IFN-2α showed an increase in the likelihood of relapse compared to the use of MMC and then BCG alone [88]. However, promising studies were published that confirmed improvement without recurrence and reduced disease progression rate using a combination of BCG and MMC with electromotive drug administration (EMDA) compared to BCG monotherapy [10,89,90].

In the case of muscle invasive and metastatic bladder cancer, the vast majority of therapies are not conducted as monotherapy, but as combined therapies. They then show much greater effectiveness and, in the case of NIMBC, reduced the chance of relapse and the risk of regression. Overall, they had a positive effect on the prognosis for patients with advanced stage cancer [85].

In reference to the EAU report from 2019 [85], several therapies can be identified that are at an advanced level of research, and some of them have also been approved for the treatment of patients with specific cases of cancer. For example, in patients entitled to receive cisplatin, a combination chemotherapy with cisplatin plus gemcitabine plus cisplatin (GC), methotrexate, vinblastine, Adriamycin plus cisplatin (MVAC) is recommended. For patients not eligible for cisplatin, checkpoint inhibitors (pembrolizumab or atezolizumab depending on PD-L1 status) are recommended. For patients who have disease progression during or after combination therapy with cisplatin, it is recommended to use checkpoint inhibitors [85].

In 2106, a meta-analysis [91] was published, and confirmed the data presented several years earlier by three other meta-analyses [92,93,94]. It showed an absolute 8% improvement in patient survival after five years with a number-needed-to-treat of 12.5 [91]. Only cisplatin combined chemotherapy with at least one additional chemotherapeutic agent brought a significant therapeutic benefit [85,92,94].

These data confirmed the speculations of scientists regarding combination therapies in the treatment of bladder cancer. The use of this type of therapy significantly improves patient survival, and also positively affects the rate of their recovery. Despite the current great successes in the treatment of bladder cancer using both monotherapy and combination therapies, further research is needed to refine existing methods and create new, possibly even more effective ones.

## 4. Ganciclovir

The initiation of apoptosis is a relatively new and extensively studied method for treating BC. An example of this type of therapy is the use of intravesical pSV-KT/GCV [54]. This therapy involves the human papillomavirus pseudovirion (pSV), the thymidine kinase (KT) suicide gene, and ganciclovir (GCV). GCV is a synthetic guanine derivative used as an antiviral drug. The vector binds to heparan sulfate, which is a polysaccharide found in the basement membrane, extracellular matrix, and surface of tumor cells. The pSV can infect tumor cells with a 10-fold greater efficacy than normal bladder cells [54,75]. When introduced into cancer cells, there is a strong overexpression of TK, which is a catalyst for GCV phosphorylation. Two more phosphorylations occur through endogenous kinases. The result is ganciclovir triphosphate, and it is incorporated into the DNA chain as a false nucleotide by the DNA polymerase. This leads to chain termination and induction of tumor cell death [54]. Tumor cell death may result in tumor-antigen presentation and T cell activation [54].

In the murine orthotopic MB49-BC model, treatment with pSV-KT in combination with GCV led to the immunogenic death of tumor cells and an increase in tumor-specific CD8 T cells in vivo. The result was tumor shrinkage and increased mouse survival [75].

## 5. The mTOR Kinase Inhibitors

The mammalian target of rapamycin (mTOR) kinase regulates many signaling pathways responsible for cell proliferation, angiogenesis, and growth. The mTOR inhibitors, such as everolimus or sirolimus, are used as immunosuppressive drugs [54].

Research on the use of these drugs to treat BC is ongoing. The drugs work by binding to the FKBP-12 (tacrolimus binding protein 12) protein, forming a complex that inhibits mTOR activity. This results in cell cycle arrest and the inhibition of angiogenesis, proliferation, and glucose delivery to cells [54]. Angiogenesis is inhibited by the reduction in the expression of hypoxia-inducible factor 1, which in turn reduces the level of vascular endothelial growth factor [54,95].

In 2016, the FDA approved everolimus for adult patients with progressive neuroendocrine tumors of gastrointestinal or lung origin with unresectable, locally advanced, or metastatic disease [96]. The efficacy of everolimus in BC treatment has been well documented both in vitro and in animal studies [54,97,98]. A phase II trial involving patients with metastatic urothelial carcinoma revealed the antitumor activity of everolimus, albeit only in a subgroup of patients [99]. In vitro studies have also shown the effectiveness of everolimus combined with cisplatin [100].

## 6. IL-12

Intravesical interleukin 12 (IL-12) stimulates the immune system and reduces the state of immunosuppression in cancer cells [54]. IL-12 is delivered along with chitosan, a mucoadhesive polysaccharide thought to increase the absorption of IL-12. Results from animal studies indicate an efficiency of 90% for this method, and also show the creation of a specific immune memory that protects against cancer recurrence [54].

IL-12 stimulates macrophages to release interferon γ leading to activation of CD3+, CD8+, and CD4+ T cells and also activates tumor-infiltrating lymphocytes. Additionally, it decreases the number of regulatory T cells and myeloid-derived suppressor cells [54,101,102].

Clinical trials using IL-12 are not yet available.

## 7. COX-2 Inhibitors

Celecoxib is a selective cyclooxygenase-2 (COX-2) inhibitor that has chemo-preventive activity against various types of cancer, including BC. It inhibits the proliferation, migration, invasion, and epithelial-to-mesenchymal transition of BC cells, but its mechanism of action is not yet fully understood [103]. The combination of celecoxib with molecules that mimic microRNA-145 showed an additive migration and invasion-inhibitory effect in BC cell lines [103].

## 8. Future Perspectives

More research is needed in this area, particularly on the profiling of gene signaling pathways, which may be the reason why some patients respond to immunotherapy and others do not. It is worth remembering that genetic instability of bladder cancer together with individual features of the patient are the keys in proper and efficient treatment. One of the novel approaches is the use of epidermal growth factor receptor (EGFR), which is a receptor with tyrosine kinase activity associated with the progression and chemoresistance of BC. The canonical role of EGFR is the regulation of epithelial tissue development and homeostasis, but in pathological settings it may be a driver of tumorigenesis. Studies have shown an over-expression of the gene encoding for EGRF and those of other proteins involved in the enzymatic cascade in 25% of patients with muscle-invasive BC [54].

Erlotinib, an EGFR inhibitor, has been shown to inhibit tumor cell proliferation in vitro. Erlotinib binds to tyrosine kinase and inhibits its activation [104]. In addition, Cetuximab, an anti-EGFR monoclonal antibody, has also been shown to be effective, although not in clinical studies [54,105,106].

The use of human epidermal growth factor receptor (HER)-2 inhibitors is also being investigated. These molecules belong to the EGFR receptor family and are pivotal in the initiation of many important signaling pathways, including MAPK (mitogen-activated protein kinase), PI3k (phosphatidylinositol-4,5-bisposphate 3-kinase), and PKC (protein kinase C) [107]. Research on a dendritic cell-based vaccine (Lapuleucel-T by Dendreon) suggests that it may be a viable treatment for patients with HER2+ urothelial BC [108]. In this treatment, peripheral blood monocytes are isolated and cultivated with a proprietary fusion protein that combines a tumor antigen with granulocyte-macrophage colony-stimulating factor. This leads to the maturation of the mononuclear cells into antigen-presenting dendritic cells and the mobilization of the immune system of the patient [108].

Also of interest is the indirect activation of the immune system by costimulatory molecules, which are known to play a role in other types of cancer. An example is OX40. This cell-surface molecule is expressed on CD4+ and CD8+ T cells and leads to their activation and recognition of a specific antigen [109]. This is thought to increase antitumor immunity and improve tumor-free survival.

It was also described [110] that using stem cells might be a promising therapeutic tool in fighting against BC, yet the variety of cancer stem cells is present and impacts the possible outcome. In a study by Jinesh et al. [111] heterogenous expression was confirmed for proteins, like, among others, myc isoforms, granzyme-B cleavage, FasL (apoptosis antigen 1 ligand), and caspase-3 and -8, leading to the conclusion, that successful therapy in BC is strongly dependent upon individual features. Thus, molecular subtyping of the type of carcinoma is crucial to gain better insight into the stem cell–cancer relationship.

Moreover, an interesting approach was suggested by Poch et al. [112] who incorporated tumor-infiltrating lymphocytes (TIL) as a method of bladder cancer treatment that is known to be a promising therapy in metastatic melanoma. Also, in response to BC, TIL proved to secrete IFN-gamma.

Interestingly, attempts were made to take advantage of a very novel and successful method of treatment in lymphoid malignancies, which is CAR (chimeric antigen receptor) T cell expression [113]. CARTs are aimed to recognize tumor-associated antigens and awaken T cell activation [113]. In the tumor microenvironment, T cells may have limited cytolytic activity and secretion due to PD-1 activation and, in order to avoid this, murine chimeric PD1 receptor (chPD1) was constructed [113]. Antitumor efficacy of chPD1 T cells was confirmed on several cell lines, including bladder cancer.

Finally, there is the matter of diversity and uniqueness of the patients involved. The best and most effective option may be a combined and personalized therapy.

## 9. Alternative Therapies to Support the Immune System in People with Bladder Cancer

The use of alternative, more natural treatment methods is popular among cancer patients seeking to boost their immune system and lower the risk of recurrence [114]. Among them, curcumin has been observed to have antitumor properties in vitro. Curcumin is a natural phenol produced by the plant *Curcuma longa*, commonly known as turmeric, and sold as an herbal supplement. Unfortunately, its clinical application has been elusive due to its low bioavailability and instability. Recently, however, curcumin coupled with light therapy was observed to improve the compound’s bioavailability in skin-related diseases [115]. In bladder cancer cell cultures, Roos et al. observed that curcumin treatment followed by light exposure had enhanced antitumor properties, inhibiting cell growth and proliferation [116].

Sulforaphane is another natural compound that has garnered attention as a complementary or alternative to traditional cancer treatments [117]. Sulforaphane is produced by cruciferous vegetables such as Brussels sprouts and cabbages and has been observed to have antitumor effects in vitro and in vivo. It is known to suppress histone deacetylases (HDACs) [117], therefore affecting the epigenetic regulation of gene expression. Up to 90% of cancers have been associated with epigenetic modifications [118]. For this reason, HDAC inhibition has been proposed as an alternative cancer treatment [117,119,120]. In the context of BC, treatment with the HDAC inhibitor valproic acid inhibited the growth and proliferation of temsirolimus-resistant BC cells [121]. Furthermore, there seems to be an inverse association between broccoli and cabbage consumption and bladder cancer risk [122,123,124]. For a more complete review on the relevance of sulforaphane in the context of BC see the review by Juengel et al. [117].

## 10. Conclusions

BC is one of the most common types of cancers and, unfortunately, has a high rate of recurrence. Treatment must often be administered multiple times, and resistance to the drugs inevitably develops. Immunotherapy is a promising alternative, and checkpoint inhibitors in particular have gradually emerged as the treatment of choice. However, not all patients respond equally to this kind of therapy and much research is still needed. Other compounds and molecular pathways are also under investigations, as well as the natural compounds present in the herbal supplements that so many patients are drawn to. As research has shown in the past, a combination of methods may be the most effective therapy.

## Figures and Tables

**Figure 1 cancers-12-01181-f001:**
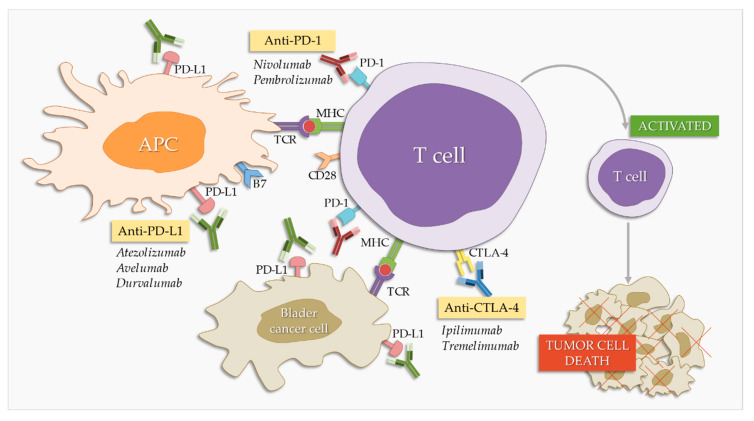
Effect of checkpoint inhibitors in bladder cancer treatment. PD-1/PD-L1 and CTLA-4 blockers interfere with the immune system’s inhibitory checkpoint molecules, leading to T cell activation and tumor cell death. APC: antigen-presenting cells.

**Table 1 cancers-12-01181-t001:** List of approved checkpoint inhibitors used in bladder cancer treatment.

Compound	Trade Name	Company	Target	Date of Approval	Clinical Trial Leading to Approval
Atezolizumab	Tecentriq	Genentech	PD-L1	2016	IMVigor210 [48]
Avelumab	Bavencio	Merck	PD-L1	2017	JAVELIN [49]
Durvalumab	Imfinzi	AstraZeneca	PD-L1	2017	Study 1108 [50]
Nivolumab	Opdivo	Bristol-Meyers Squibb	PD-1	2017	CheckMate 275 [51]
Pembrolizumab	Keytruda	Merck	PD-1	2019	KEYNOTE-057 [52]
Ipilimumab	Yervoy	Bristol-Meyers Squibb	CTLA-4	2019	NCT01524991 [53]

PD-L1: programmed death ligand 1; PD-1: programmed cell death protein 1; CTLA-4: Cytotoxic T lymphocyte-associated protein 4.

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
