# Peer review of "Immunotherapy in Bladder Cancer: Current Methods and Future Perspectives"

_cancers, 2020, doi:10.3390/cancers12051181_

Round 1
Reviewer 1 Report
This is a very well written review. It could be improved by attention to a few details:
1.I think it would be worthwhile explaining the importance of genetic instability of the tumor cell, which results in the generation of neoantigens, for the efficacy of checkpoint inhibitors. Some cancers, colorectal for example, are not responsive to checkpoint inhibitors because, in general, they are genetically stable. What is the situation for bladder cancer? This is alluded to on line 141 regarding the improved success of these inhibitor in bladder cancer in smokers, but never really explained. It could come in the introduction section 3, on page 2.,
2. I think section 3, would best be titled "Checkpoint inhibitors" and include a description of the anti-CTLA4 antibodies as well as the PD-1/.PD-L1pathway.
3. In section 5, line 191. You should include a reference for the statement "Tumor cell death may result in tumor-antigen presentation and T cell activation"
4. Section 7, line 210. I would not call IL-12 an inhibitor, it is a lymphokine, an immune stimulator. Also on line 216, IL-12 does not "infiltrate" T cells. I don't think infiltrates is the correct word. IL-12 stimulates macrophages to release interferon-gamma, which activates T cells.
5. Section 9, line 228. It might also be worth mentioning here that individual gets instability of the tumor plays an important role in why some patients respond to checkpoint inhibitors than others.
6. Section 9, should include the potential of cell therapies for bladder cancer. Although, early CAR T cells are being explored for the therapy of bladder cancer.
Author Response
Dear Reviewer,
On behalf of the Authors of the paper entitled “Immunotherapy in Bladder Cancer – Current Methods and Future Perspectives”, we would like to cordially thank for the instructive and detailed review of our paper. We believe that the your outstanding knowledge and involvement impacted our paper and made it much better. We followed your suggestions and tried to fulfill all of your hints and all changes are marked in red in the updated version of the manuscript. Here are the point-by point answers.
- I think it would be worthwhile explaining the importance of genetic instability of the tumor cell, which results in the generation of neoantigens, for the efficacy of checkpoint inhibitors. Some cancers, colorectal for example, are not responsive to checkpoint inhibitors because, in general, they are genetically stable. What is the situation for bladder cancer? This is alluded to on line 141 regarding the improved success of these inhibitor in bladder cancer in smokers, but never really explained. It could come in the introduction section 3, on page 2.
* We added a paragraph (in the place of the manuscript suggested by the Reviewer) on the genetic instability of the bladder cancer, to try to draw the Readers attention to the fact, that this factor may cause differences in the efficacy of checkpoint inhibitors.
- I think section 3, would best be titled "Checkpoint inhibitors" and include a description of the anti-CTLA4 antibodies as well as the PD-1/.PD-L1pathway.
* We changed the section title according to the Reviewer suggestion and included PD-1/PD-L1 and CLTA-4 in it.
- In section 5, line 191. You should include a reference for the statement "Tumor cell death may result in tumor-antigen presentation and T cell activation"
* The reference was added
- Section 7, line 210. I would not call IL-12 an inhibitor, it is a lymphokine, an immune stimulator. Also on line 216, IL-12 does not "infiltrate" T cells. I don't think infiltrates is the correct word. IL-12 stimulates macrophages to release interferon-gamma, which activates T cells.
* we changed the title of the section to avoid calling IL-12 an inhibitor and rephrased the sentence with the misleading word “infiltrate”
- Section 9, line 228. It might also be worth mentioning here that individual gets instability of the tumor plays an important role in why some patients respond to checkpoint inhibitors than others.
* we added the information of the genetic instability of BC playing a pivotal role in the responsiveness to the checkpoint inhibitors’ treatment
- Section 9, should include the potential of cell therapies for bladder cancer. Although, early CAR T cells are being explored for the therapy of bladder cancer.
- we expanded the section with the CAR T therapy, stem cell and TIL treatment.
Again, we would like to thank you for your effort and time and we are hoping that our manuscript in its current for will fulfill the requirements of the Cancer journal.
Thank you for your time and consideration,
Ewelina Grywalska & Paulina NiedĹşwiedzka-Rystwej
Reviewer 2 Report
In this review article, Mikołaj Wołącewicz and colleagues try to provide the current methods and future perspectives of immunotherapy. Unfortunately, this review is not updated and some issues are event out of clinical practice. Also, this review is not written logically. There are some suggestions for this manuscript:
- In the introduction, the lack of logically introduce the rationale of immunotherapy. This section starts with bladder cancer, followed by the risk factors, therapy and prompt to immunotherapy.
- In the section of BCG, there are lots of studies focus on this issue in recent years. The references and information are not updated.
- For the immune checkpoint inhibitors, the references and information are not updated. Even, some sections just discuss to the phase II trial. This is not enough for high impact review. Today, combination therapy and the sequential use of immune checkpoint inhibitors and other therapies are important issues in the real world. None of them is reviewed.
- Also, there are some medications that not included in the current EAU guideline. These issues should be reviewed carefully and, the same concern, the references are not updated.
Author Response
Dear Reviewer,
On behalf of the Authors of the paper entitled “Immunotherapy in Bladder Cancer – Current Methods and Future Perspectives”, we would like to cordially thank for the instructive and detailed reviews of our paper. We believe that your outstanding knowledge and involvement, impacted our paper and made it much better. We followed all your suggestions and tried to fulfill all of your hints and all changes are marked in red in the updated version of the manuscript. Here are the point-by point answers.
Reviewer 2
- In the introduction, the lack of logically introduce the rationale of immunotherapy. This section starts with bladder cancer, followed by the risk factors, therapy and prompt to immunotherapy.
*we expanded the Introduction with information on immunotherapy, to (hopefully) increase the logic that was missing
- In the section of BCG, there are lots of studies focus on this issue in recent years. The references and information are not updated.
*The references and information have been updated.
- For the immune checkpoint inhibitors, the references and information are not updated. Even, some sections just discuss to the phase II trial. This is not enough for high impact review. Today, combination therapy and the sequential use of immune checkpoint inhibitors and other therapies are important issues in the real world. None of them is reviewed.
*We added additional section on combined therapies to match the Reviewer expectations and to show the full image of bladder cancer treatment.
- Also, there are some medications that not included in the current EAU guideline. These issues should be reviewed carefully and, the same concern, the references are not updated.
*We consulted the EAU guidelines and included the information in the text, which of the drug is recommended by FDA, and EAU. The references have been updated.
Again, we would like to thank you for your effort and time and we are hoping that our manuscript in its current for will fulfill the requirements of the Cancer journal.
Thank you for your time and consideration,
Ewelina Grywalska & Paulina NiedĹşwiedzka-Rystwej
Reviewer 3 Report
The Review article by Wolacewicz et al., "Immunotherapy in Bladder Cancer - Current Methods and Future Perspectives" provides an overview of therapies for bladder cancer that relate to infiltrating immune cells. The study summarizes approved immune checkpoint treatments and follows with other treatments that work as independent therapies or having potential to work as combination in order to increase ICI efficiency. While the Review provides little new information or innovation, it is well written, easy to read and provides the essential treatment information for those interested in urothelial immunotherapies.
The Abstract, however, needs to be re written to include the main ideas of the Review in greater detail and written in better English. Statements such as, "with the development of molecular biology....", "Research suggest that the combination of several methods..." are vague and need rephrasing.
Author Response
Dear Reviewer,
On behalf of the Authors of the paper entitled “Immunotherapy in Bladder Cancer – Current Methods and Future Perspectives”, we would like to cordially thank for the instructive and detailed review of our paper. We believe that your outstanding knowledge and involvement, impacted our paper and made it much better. We followed your instructions and changes are marked in red in the updated version of the manuscript. Here are the point-by point answers.
Reviewer 3
- The Abstract, however, needs to be re written to include the main ideas of the Review in greater detail and written in better English. Statements such as, "with the development of molecular biology....", "Research suggest that the combination of several methods..." are vague and need rephrasing.
*The Abstract has been rewritten, to avoid vague sentences and to make it more informative, as the Reviewer suggested.
Again, we would like to thank you for your effort and time and we are hoping that our manuscript in its current for will fulfill the requirements of the Cancer journal.
Thank you for your time and consideration,
Ewelina Grywalska & Paulina NiedĹşwiedzka-Rystwej
Round 2
Reviewer 2 Report
The authors answered all question accordingly.